# Suboptimal Performance of Hepatocellular Carcinoma Prediction Models in Patients with Hepatitis B Virus-Related Cirrhosis

**DOI:** 10.3390/diagnostics13010003

**Published:** 2022-12-20

**Authors:** Jae Seung Lee, Tae Seop Lim, Hye Won Lee, Seung Up Kim, Jun Yong Park, Do Young Kim, Sang Hoon Ahn, Hyun Woong Lee, Jung Il Lee, Ja Kyung Kim, In Kyung Min, Beom Kyung Kim

**Affiliations:** 1Department of Internal Medicine, Yonsei University College of Medicine, Seoul 03722, Republic of Korea; 2Institute of Gastroenterology, Yonsei University College of Medicine, Seoul 03722, Republic of Korea; 3Yonsei Liver Center, Severance Hospital, Seoul 03722, Republic of Korea; 4Division of Gastroenterology, Department of Internal Medicine, Yongin Severance Hospital, Yonsei University Health System, Gyeonggi-do, Seoul 03722, Republic of Korea; 5Division of Gastroenterology, Department of Internal Medicine, Gangnam Severance Hospital, Yonsei University Health System, Seoul 06273, Republic of Korea; 6Biostatistics Collaboration Unit, Department of Biomedical Systems Informatics, Yonsei University College of Medicine, Seoul 03722, Republic of Korea

**Keywords:** chronic hepatitis B, liver cirrhosis, antiviral therapy, hepatocellular carcinoma, risk, liver stiffness

## Abstract

This study aimed to evaluate the predictive performance of pre-existing well-validated hepatocellular carcinoma (HCC) prediction models, established in patients with HBV-related cirrhosis who started potent antiviral therapy (AVT). We retrospectively reviewed the cases of 1339 treatment-naïve patients with HBV-related cirrhosis who started AVT (median period, 56.8 months). The scores of the pre-existing HCC risk prediction models were calculated at the time of AVT initiation. HCC developed in 211 patients (15.1%), and the cumulative probability of HCC development at 5 years was 14.6%. Multivariate Cox regression analysis revealed that older age (adjusted hazard ratio [aHR], 1.023), lower platelet count (aHR, 0.997), lower serum albumin level (aHR, 0.578), and greater LS value (aHR, 1.012) were associated with HCC development. Harrell’s c-indices of the PAGE-B, modified PAGE-B, modified REACH-B, CAMD, aMAP, HCC-RESCUE, AASL-HCC, Toronto HCC Risk Index, PLAN-B, APA-B, CAGE-B, and SAGE-B models were suboptimal in patients with HBV-related cirrhosis, ranging from 0.565 to 0.667. Nevertheless, almost all patients were well stratified into low-, intermediate-, or high-risk groups according to each model (all log-rank *p* < 0.05), except for HCC-RESCUE (*p* = 0.080). Since all low-risk patients had cirrhosis at baseline, they had unneglectable cumulative incidence of HCC development (5-year incidence, 4.9–7.5%). Pre-existing risk prediction models for patients with chronic hepatitis B showed suboptimal predictive performances for the assessment of HCC development in patients with HBV-related cirrhosis.

## 1. Introduction

Hepatitis B virus (HBV) infection remains the leading etiology of hepatocellular carcinoma (HCC) globally and cirrhosis. The current mainstay of treatment for chronic hepatitis B (CHB) is long-term antiviral therapy (AVT) using potent oral nucleos(t)ide analogs (NUCs), e.g., entecavir (ETV) or tenofovir disoproxil fumarate (TDF), that effectively suppress the replication of HBV DNA and decrease the risk of HCC development [1,2,3]. However, periodic surveillance to detect HCC which allows curative approaches is still mandatory for patients with CHB [4,5,6,7,8]. This is because AVT does not completely eliminate the risk of HCC development [9,10,11,12].

Many models have been suggested to assess the risk stratification of HCC development in CHB patients [13,14,15]. Since the prognostic role of serum HBV DNA has weakened in the current era of potent NUCs, models established within one decade have adopted the presence of baseline cirrhosis and/or fibrosis parameters, rather than virological factors such as hepatitis B e-antigen (HBeAg) and/or serum HBV-DNA level [16,17,18,19,20,21,22,23]. This provides an overall superior prognostic performance to old models (i.e., REVEAL [24], CU-HCC [25], GAG-HCC [26], LSM-HCC [27], and REACH-B [28]), which primarily depend on virological factors.

However, although cirrhotic patients are more likely to develop HCC by up to more than 10 times compared to non-cirrhotic patients, there is an unmet need to develop optimized models that allow for earlier intervention. However, no study has assessed the performance of recently validated HCC risk prediction in such a population. Since HCC prediction models so far have generally incorporated cirrhosis itself, or surrogate markers suggestive of cirrhosis, as major integral components, most of which were based on routine ultrasonography, clinical parameters, and non-invasive fibrosis measurements, it remains undetermined as to whether the reliable predictive performances might be maintained among a population with HBV cirrhosis.

Therefore, using a cohort with HBV-related cirrhosis, we aimed to evaluate the predictive performance of pre-existing well-validated HCC prediction models established in the era of potent AVT.

## 2. Materials and Methods

### 2.1. Study Design and Participants

Patients with cirrhosis, who initiated ETV or TDF as the first-line AVT for treatment-naïve CHB between 2007 and 2018 at Yonsei University Severance Hospital, Gangnam Severance Hospital, and Yongin Severance Hospital, were retrospectively reviewed. The inclusion criteria were as follows: (1) adult patients with age ≥ 19 years, (2) who were AVT-naïve, and (3) with reliable baseline liver stiffness (LS) value measured using transient elastography (TE). The exclusion criteria were as follows: (1) without having cirrhosis, (2) history of HCC at enrollment, (3) decompensated cirrhosis with Child–Pugh class C at enrollment, (4) co-infection with other hepatitis viruses or human immunodeficiency virus, (5) history of organ transplant, (6) HCC development within 6 months of AVT initiation, and (7) other significant comorbidities (e.g., end-stage kidney disease, uncontrolled heart failure, pulmonary hypertension, and life-threatening autoimmune disease) (Appendix A). AVT was initiated according to the practice guidelines of the Korean Association for the Study of the Liver and the reimbursement guidelines of the National Health Insurance Service of the Republic of Korea (ROK). Cirrhosis was diagnosed histologically or clinically as follows: (1) with a platelet count <150,000/μL and ultrasonographic findings suggestive of cirrhosis, including a blunted, nodular liver edge accompanied by splenomegaly (length > 12 cm), or (2) with clinical signs of portal hypertension such as gastroesophageal varices [29].

The study protocol was in accordance with the ethical guidelines of the 1975 Declaration of Helsinki and was approved by the institutional review board in each medical center.

### 2.2. HCC Surveillance

Patients underwent routine laboratory testing assays of serum levels of HBV-DNA, as well as liver imaging studies (e.g., ultrasonography or computed tomography) at approximately 6-month intervals after initiating AVT to screen for HCC and portal hypertension-related complications. LS was measured using TE (FibroScan^®^, EchoSens, Paris, France), and was considered to be reliable when the procedure was performed with at least 10 valid measurements, a success rate of at least 60%, and an interquartile range (IQR)-to-median ratio of <30% in a standard manner [30].

The primary outcome was the development of HCC. HCC was diagnosed based on histological evidence or dynamic computed tomography and/or magnetic resonance imaging findings (nodules > 1 cm with arterial hypervascularity and portal-/delayed-phase washout) [31].

### 2.3. Calculation of HCC Risk Scores from Prediction Models

The scores of pre-existing HCC risk prediction models were calculated at the time of AVT initiation to predict HCC development after 6 months of AVT use. These models included PAGE-B [16], modified PAGE-B [17], modified REACH-B [18], CAMD [19], aMAP [32], Toronto HCC Risk Index (THRI) [33], AASL-HCC [14], HCC-RESCUE [34], PLAN-B [35], and APA-B (in patients with alpha-fetoprotein [AFP] results) [36]. In general, CAGE-B and SAGE-B are calculated using the LS value, stabilized after 5 years of AVT [20,21]. However, considering that the LS value significantly improves after 1 year of AVT [37], CAGE-B and SAGE-B scores were also calculated after, using the LS value in the patient group with follow-up TE results after 1 year of AVT, and their performances were compared with other models. Therefore, CAGE-B and SAGE-B were calculated at the time of AVT initiation to predict HCC development after 18 months of AVT use. The list of these models and the risk stratification are summarized in Appendix A. Patients were stratified into the low-, intermediate-, and high-risk groups according to the previous studies that introduced each prediction model [14,16,17,18,19,20,21,32,33,34,35,36].

### 2.4. Statistical Analysis

Continuous variables were expressed as medians (IQRs), and categorical variables were expressed as numbers (percentages). The statistical differences between the two groups were evaluated using Student’s *t* test or the Mann–Whitney U test for continuous variables, and using the chi-squared test or Fisher’s exact probability test, respectively, depending on their distribution. The cumulative risk of HCC development was assessed by the Kaplan–Meier method. Patients were censored from the results when they ended the follow-up, died without developing HCC, or developed other malignant diseases rather than HCC. Univariate and subsequent multivariate Cox regression analyses assessed the potential risk factors and their independent associations for HCC development, respectively, by calculating the hazard ratio (HR) and 95% confidence interval (CI).

The predictive performance of the risk scoring models for HCC development was assessed using Harrell’s C-indices, time-dependent areas under the receiver operating characteristic curve (TDAUCs) at 3, 5, and 8 years from the date initiating AVT, and the integrated area under the receiver operating characteristic curve (iAUC) after 8 years. These were chosen because there were few patients who followed up for >8 years after initiating AVT. Statistical differences in the parameters for predictive performances between the model with highest iAUC and other HCC risk prediction models were evaluated using the bootstrap method, with re-sampling done 1000 times. If the 95% CI contains zero, there is no significant difference in parameters for predictive performances between two models.

To calculate the PLAN-B model, we used Python programming language (version 3.11; Python Software Foundation, Wilmington, DE, USA) and assessed the shared source code that is available online at https://github.com/vitaldb/planb/blob/main/predict.ipynb (accessed on 25 November 2022) [35]. All statistical analyses were conducted using R software (version 4.2.1, http://cran.r-project.org/) (accessed on 15 August 2022). Two-sided *p* values < 0.05 were considered to be statistically significant.

## 3. Results

### 3.1. Baseline Characteristics and HCC Development

According to the enrollment criteria, 1399 treatment-naïve cirrhotic patients with CHB were recruited (Appendix A). The median age was 54.0 (interquartile range [IQR], 47.0–59.0) years, with a male predominance of 53.5%. Tenofovir was initiated in 684 (51.1%) patients. HBeAg positivity was detected in 514 (38.4%) patients. TE at baseline and 1 year after AVT (*n* = 808) revealed median LS of 11.2 (IQR, 7.4–17.3) kPa and 8.9 (2.7–13.4) kPa, respectively (Table 1).

During a median follow-up period of 56.8 (IQR 35.6–75.3) months, HCC developed in 211 (15.1%) patients (3.41 per 100 patient-years) and the cumulative 3-, 5-, and 8-year probabilities of HCC development were 7.4%, 14.6%, and 31.7%, respectively. Patients who developed HCC showed significantly older age (55 vs. 53 years); higher HBeAg positivity (47.4% vs. 36.7%); lower platelet count and serum albumin level; and higher values of baseline and follow-up LS (14.3 vs. 10.3 kPa, and 11.8 vs. 8.7 kPa, respectively), compared to those without HCC (Table 2). The median scores for the pre-existing predictive models for HCC development were significantly higher in patients who developed HCC than in those who did not (Table 2).

### 3.2. Independent Predictive Factors of HCC Development in Cirrhotic Patients with CHB

Univariate Cox regression analysis revealed that age, the presence of diabetes mellitus, HBeAg positivity, lower platelet counts, lower serum albumin levels, and greater LS values were significantly associated with HCC development (Appendix A). Subsequent multivariable analysis revealed that older age (aHR, 1.023; 95% CI, 1.008–1.038), lower platelet count (aHR, 0.997; 95% CI, 0.994–0.999), lower serum albumin level (aHR, 0.578; 95% CI, 0.446–0.751), and greater LS (aHR, 1.012; 95% CI, 1.002–1.024) were independently associated with an increased risk of HCC development (Table 3).

### 3.3. Predictive Performance and HCC

The Harrell’s c-index, iAUC, and the 1-, 2-, 3-, 5-, and 8-year TDAUCs, were summarized in Table 4. Among the prediction models using baseline variables, modified REACH-B showed the highest c-index (0.667) and iAUC (0.643). However, their values did not reach an acceptable level (<0.7). The modified REACH-B showed significantly higher iAUC than other risk models which used the bootstrap resampling method, except for those of PLAN-B (−0.030, 95% CI −0.066–0.006), APA-B (−0.011, 95% CI −0.055–0.034, *n* = 910), CAGE-B (−0.034, 95% CI −0.076–0.007, *n* = 808), and SAGE-B (−0.020, 95% CI −0.063–0.022, *n* = 808) (Appendix A). CAGE-B showed significantly lower iAUC compared to those of SAGE-B (0.622 vs. 0.639, 0.014 [95% CI, 0.002–0.027], *n* = 808).

### 3.4. Risk Stratification in Cirrhotic Patients with CHB

Patients were stratified into low-, intermediate-, and high-risk groups according to the models, which showed that the risk of HCC development increased in the high-risk group of each model (all log-rank *p* < 0.05) (Figure 1). There were more than 10% of patients who stratified into the low-risk group according to the modified REACH-B, PLAN-B, APA-B, and SAGE-B (13.8–24.5%), and the risk was significantly or tended to be lower than that in the intermediate- and high-risk groups (all log-rank *p* < 0.05, except for APA-B [*p* = 0.050]). However, these patients also showed a high cumulative incidence of HCC (5-year incidence, 4.9%–7.5%), even when stratified into the low-risk group (Table 5).

### 3.5. On-Treatment LS Value in Cirrhotic Patients with CHB

The baseline characteristics of patients who had TE data after 1 year of AVT and did not develop HCC within 18 months after AVT (*n* = 808) are summarized in Appendix A. The median value of on-treatment LS was 8.8 kPa. Patients with an on-treatment LS value ≥8.8 kPa had a higher risk of HCC development than the others (unadjusted hazard ratio = 2.252, 95% CI, 1.500–3.383, *p* < 0.001). The 2-, 3-, 5-, and 8-year cumulative incidences of HCC development were 1.6%, 3.6%, 11.0%, and 23.7% in patients with on-treatment LS value <8.8 kPa, respectively, and 3.6%, 10.5%, 19.9%, and 56.0% in patients with on-treatment LS value ≥8.8 kPa, respectively (log-rank *p* < 0.001).

## 4. Discussion

To date, several risk-scoring systems have been proposed to predict the development of HCC in patients with CHB. In the current era of potent AVT where the virologic effects can be easily suppressed, most of the recently established systems adopted the presence of baseline cirrhosis or fibrotic burden, and generally demonstrated high negative predictive values to exclude HCC development within about 10 years [38]. However, because cirrhosis itself is a strong predictor [39], the predictive power of the proposed models is expected to decrease somewhat in the cirrhosis group, which has a common fibrotic burden [40].

In the present study, age, platelet count, serum albumin level, HBeAg positivity, and LS value remained independent or tended to be associated with HCC development in patients with HBV-related cirrhosis. However, regardless of the presence of cirrhosis as a component in the scoring system, several of the models introduced, partially based on these factors, showed attenuated predictive performance for HCC development in the subgroup with HBV cirrhosis (all Harrell’s c-index and iAUC < 0.7). These findings are similar to those of previous studies that have attempted to develop predictive models for patients with HBV cirrhosis. Cheng et al. [41] reported that the predictive performance of CU-HCC, PAGE-B, modified PAGE-B, and their suggested HCC-nomogram using albumin-bilirubin score at 1-year of AVT in 277 treatment-naïve patients with HBV cirrhosis was very limited (0.505–0.611). Nam et al. [42] also reported that the PAGE-B, CU-HCC, HCC-RESCUE, ADRESS-HCC, mPAGE-B, and THRI models showed very poor performance (c-index of all models < 0.6) in 424 patients, compared to that of their suggested deep neural network model (c-index: 0.782). Huang et al. [43] contrary demonstrated that the GAG-HCC, REACH-B, and TW1 models showed acceptable AUCs (0.747–0.797) by 5 years after AVT, however, the study might be insufficient to reflect the realities of the current era due to the relatively small number of participants (*n* = 226) who were treated with lamivudine or adefovir.

Patients with HBV cirrhosis have a higher risk of HCC than those without cirrhosis [39]. Since most of the patients in our study were clinically diagnosed with cirrhosis using ultrasonography and clinical parameters, there might be higher possibilities of the over-estimation of cirrhosis. Since most of the patients in our study were clinically diagnosed with cirrhosis using ultrasonography and clinical parameters, there might be higher possibilities of over-estimation of cirrhosis, when compared to diagnosis by non-invasive fibrosis tests, such as TE, Fibrotest, or the enhanced fibrosis test [44]. However, most participants were stratified into moderate- or high-risk groups by most scoring systems. Therefore, the reported annual incidence of HCC at 3.41 per 100 patient-years was higher than the recommended criteria for the biannual HCC surveillance strategy (≥1.5% in cirrhosis) [45]. Moreover, even though patients were sufficiently (>10% of total) classified as low-risk by the models that did not have cirrhosis components in their equations (e.g., modified REACH-B, APA-B, and SAGE-B), they showed a non-negligible 5-year cumulative incidence of HCC (6.7–7.5%). This was quite different from the previously reported 5-year cumulative HCC incidences (<1.0%) in patients with CHB, regardless of the presence of cirrhosis. Even patients with an LS value that improved to less than 8.8 kPa after 1 year of AVT also showed a high 5-year cumulative incidence rate (11.0%). These findings indicate that the candidates needing HCC surveillance, along with the optimal methods in terms of diagnostic modalities and/or interval among the so-called “at-risk” population, should not be determined solely based upon HCC prediction models.

In the present study, modified REACH-B, using the LS component, showed significantly or tended to have higher c-index and iAUC than the other models. However, the model using LS value (modified REACH-B, CAGE-B, and SAGE-B) did not continuously show the higher TDAUCs at 1, 2, and 3 years after AVT initiation. Considering that patients with liver cirrhosis are at risk of developing HCC, even within a relatively short period of time after follow-up, the superiority of the model cannot be quickly determined. This is the case even if the c-index or integrated AUC is high in the modified REACH-B.

Notably, multivariate Cox regression analyses revealed that the known risk factors for HCC development in patients with CHB on AVT, such as old age, low platelet count, low serum albumin level, and high LS value by TE [22], were still independently associated with HCC development in patients with HBV cirrhosis. Moreover, patients who showed a very high cumulative incidence of HCC development were classified as a high-risk group by the models containing all or some of these risk factors, such as modified PAGE-B, modified REACH-B, CAGE-B, and SAGE-B (5-year: 15.5–24.0%, and 8-year: 42.4–52.8%). Therefore, even though cirrhosis itself can degrade the discriminating power of the variables constituting the existing predictive models in the HBV-related cirrhosis group, patients with cirrhosis, who are older, have low platelet counts, or show high LS values, should undergo stricter surveillance for HCC development, compared to those who without cirrhosis.

The present study has several limitations. First, the findings were potentially subject to selection bias owing to the retrospective nature of the study. To overcome this limitation, the study was conducted using three tertiary referral hospital-based cohorts with a statistically reliable sample size and follow-up duration. Second, since we primarily adopted the diagnostic criteria of cirrhosis based upon the ultrasonography findings and platelet count, a significant number of mild cases had been missed. Conversely, some of enrolled patients had a low LS value, despite being diagnosed using the above criteria. Thus, another kind of potential selection bias might occur. Further studies, based upon the more accurate diagnostic modalities, are required to overcome this issue. Third, this study did not suggest a novel risk model for HCC development in patients with HBV cirrhosis. A recently proposed deep learning model, using previously known risk factors, showed acceptable predictive power for HCC development in patients with HBV cirrhosis (c-index, 0.719–0.782); however, it did not represent an intuitive formula [42]. Fourth, the evaluation of new biomarkers for chronic HBV infection (e.g., quantitative HBV surface antigen, serum HBV RNA, hepatitis B core-related antigen, or specific HBV mutants) was limited because of the retrospective nature of our study [46,47,48]. Likewise, the role of other metabolic factors should be assessed in the further studies [11,12,49,50]. Finally, the present study cannot clarify whether this phenomenon is specific to the HBV or also present in the other etiologies. However, theoretically, since “cirrhosis” itself had been emphasized as one of the most important prognostic factors in most HCC prediction models so far, and its discriminatory ability must be statistically offset in the cohort with cirrhosis, we cautiously speculate that a similar phenomenon might be observed in patients with other chronic liver diseases. Further studies are required to address this issue.

## 5. Conclusions

In conclusion, the existing risk prediction models for patients with CHB showed suboptimal predictive performances for assessing HCC development in patients with HBV cirrhosis. These cirrhotic patients with CHB should undergo strict HCC surveillance, regardless of whether they have known risk factors for HCC development.

## Figures and Tables

**Figure 1 diagnostics-13-00003-f001:**
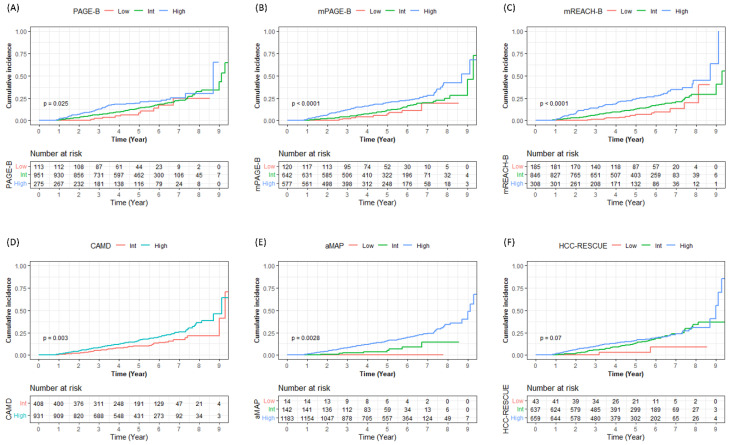
Risk stratification of HCC development according to the risk group of each PAGE-B (**A**), mPAGE-B (**B**), mREACH-B (**C**), CAMD (**D**), aMAP (**E**), HCC-RESCUE (**F**), AASL-HCC (**G**), THRI (**H**), PLAN-B (**I**), APA-B (*n* = 910) ^†^ (**J**), CAGE-B (*n* = 808) ^‡^ (**K**), and SAGE-B (*n* = 808) ^‡^ (**L**) model. ^†^ APA-B were calculated for HCC development after 6 months in 910 patients with baseline alpha-fetoprotein result. ^‡^ CAGE-B and SAGE-B were calculated for HCC development after 18 months in 808 patients, with follow-up transient elastography results after 12 months. Abbreviation: HCC, hepatocellular carcinoma; Low, low-risk group; Int, intermediate-risk group; High, high-risk group; mPAGE-B, modified PAGE-B; mREACH-B, modified REACH-B; THRI, Toronto HCC Risk Index.

**Table 1 diagnostics-13-00003-t001:** Baseline clinical characteristics of the study population.

Variables	Total (*n* = 1339)
Age (year)	54 (47, 59)
<40	87 (6.5)
40–50	343 (25.6)
50–60	578 (43.2)
60–70	265 (19.8)
≥70	51 (3.8)
Male sex	717 (53.5)
Diabetes mellitus	190 (14.2)
HBeAg positivity	514 (38.4)
TDF use (vs. ETV)	684 (51.1)
Platelet count (×10^3^/μL)	134 (99, 172)
Total bilirubin (mg/dL)	0.9 (0.7, 1.3)
Serum albumin (g/dL)	4.2 (3.8, 4.4)
Prothrombin time (INR)	1.04 (0.98, 1.13)
Aspartate aminotransferase (IU/L)	39 (28, 59)
Alanine aminotransferase (IU/L)	37 (25, 59)
Alpha-fetoprotein (ng/mL) (*n* = 910)	4.48 (2.74, 8.57)
Liver stiffness value ^†^ (kPa)	11.2 (7.4, 17.3)
1 year after AVT (kPa) (*n* = 808)	8.9 (2.7, 13.4)
Follow-up and treatment duration (month)	56.8 (35.6, 75.3)
PAGE-B	15 (12, 18)
Modified PAGE-B	12 (10, 14)
Modified REACH-B	9 (8, 11)
CAMD	15 (13, 16)
aMAP	67.3 (63.0, 71.3)
HCC-RESCUE	84 (77, 92)
AASL-HCC	19 (17, 20)
Toronto HCC Risk Index	236 (197, 297)
PLAN-B	0.395 (0.306, 0.493)
APA-B (*n* = 910)	6 (3, 8)
CAGE-B (*n* = 808)	9 (8, 11)
SAGE-B (*n* = 808)	6 (6, 9)

Values are expressed as a *n* (%) or median (interquartile range). ^†^ Measured using transient elastography (FibroScan^®^, EchoSens, Paris, France). Abbreviation: TDF, tenofovir disoproxil fumarate; ETV, entecavir; HBeAg, hepatitis B e antigen; AVT, antiviral therapy; AST, aspartate aminotransferase; ALT, alanine aminotransferase; INR, international normalized ratio.

**Table 2 diagnostics-13-00003-t002:** Comparison of baseline clinical characteristics between patients with HCC and without.

Variables	Without HCC (*n* = 1128)	HCC (*n* = 211)	*p* Value
Age (year)	53 (47, 59)	55 (50, 60)	0.003
<40	83 (7.4)	4 (1.9)	0.004
40–50	295 (26.2)	48 (22.7)	
50–60	473 (41.9)	105 (49.8)	
60–70	220 (19.5)	45 (21.3)	
≥70	42 (3.7)	9 (4.3)	
Male sex	605 (53.6)	112 (53.1)	0.942
Diabetes mellitus	151 (13.4)	39 (18.5)	0.066
HBeAg positivity	414 (36.7)	100 (47.4)	0.004
TDF use (vs. ETV)	576 (51.1)	108 (51.2)	>0.999
Platelet count (×10^3^/μL)	138 (102, 175)	115 (87, 156)	<0.001
Total bilirubin (mg/dL)	0.9 (0.7, 1.2)	1.0 (0.7, 1.5)	0.065
Serum albumin (g/dL)	4.2 (3.9, 4.4)	4.0 (3.4, 4.3)	<0.001
Prothrombin time (INR)	1.04 (0.98, 1.11)	1.05 (1.0, 1.16)	0.007
Aspartate aminotransferase (IU/L)	38 (27, 56)	49 (37, 76)	<0.001
Alanine aminotransferase (IU/L)	36 (24, 56)	44 (30, 78)	0.218
Alpha-fetoprotein (ng/mL)	3.96 (2.59, 7.46)	7.21 (4.66, 14.69)	<0.001
Liver stiffness value ^†^ (kPa)	10.3 (6.9, 16.6)	14.2 (10.0, 22.3)	<0.001
1 year after AVT (kPa) (*n* = 808)	8.7 (6.1, 12.5)	11.8 (8.65, 16.6)	<0.001
Follow-up and treatment duration (month)	60.1 (38.1, 76.1)	40.7 (24.6, 60.9)	<0.001
PAGE-B	15 (12, 18)	16 (13, 18)	<0.001
Modified PAGE-B	12 (10, 14)	13 (11, 15)	<0.001
Modified REACH-B	9 (7, 11)	11 (9, 12)	<0.001
CAMD	14 (13, 16)	15 (14, 16)	0.022
aMAP	66.8 (62.6, 71.1)	69.4 (66.1, 72.6)	<0.001
HCC-RESCUE	84 (76, 92)	86 (79, 93)	0.013
AASL-HCC	19 (17, 20)	20 (17, 22)	<0.001
Toronto HCC Risk Index	236 (197, 297)	247 (217, 297)	0.001
PLAN-B	0.382 (0.294, 0.479)	0.434 (0.370, 0.527)	<0.001
APA-B (*n* = 910)	5 (3, 8)	7 (6, 10)	<0.001
CAGE-B (*n* = 808)	9 (7, 11)	11 (9, 12)	<0.001
SAGE-B (*n* = 808)	6 (4, 9)	8 (6, 11)	<0.001

Values are expressed as a *n* (%) or median (interquartile range). ^†^ Measured using transient elastography (FibroScan^®^, EchoSens, Paris, France). Abbreviation: HBeAg, hepatitis B e antigen; TDF, tenofovir disoproxil fumarate; ETV, entecavir; AVT, antiviral therapy; AST, aspartate aminotransferase; ALT, alanine aminotransferase; INR, international normalized ratio.

**Table 3 diagnostics-13-00003-t003:** Multivariate Cox regression analysis for the development of hepatocellular carcinoma.

Variable	Univariate	Multivariate Analysis	
	*p* Value	*p* Value	Hazard Ratio (95% CI)
Age (year)	<0.001	0.003	1.023 (1.008, 1.038)
Diabetes mellitus	0.032	0.207	1.259 (0.881, 1.800)
HBeAg positivity	0.012	0.066	1.302 (0.982, 1.725)
Platelet count (×10^3^/μL)	<0.001	0.015	0.997 (0.994, 0.999)
Total bilirubin (mg/dL)	0.048	0.439	0.971 (0.901, 1.046)
Serum albumin (g/dL)	<0.001	<0.001	0.578 (0.446, 0.751)
Prothrombin time (INR)	0.014	0.589	0.829 (0.420, 1.636)
Liver stiffness value ^†^ (kPa)	<0.001	0.026	1.012 (1.002, 1.024)

^†^ Measured using transient elastography (FibroScan^®^, EchoSens, Paris, France). Abbreviation: HBeAg, hepatitis B e-antigen; INR, international normalized ratio.

**Table 4 diagnostics-13-00003-t004:** Predictive performance of the risk prediction models.

Scoring Systems	Harrell’s c-Index (95% CI)	Integrated AUC * (95% CI)	TDAUC at 1 Year (95% CI)	TDAUC at 2 Year (95% CI)	TDAUC at 3 Year (95% CI)	TDAUC at 5 Year (95% CI)	TDAUC at 8 Year (95% CI)
PAGE-B	0.605 (0.568, 0.644)	0.573 (0.536, 0.609)	0.674 (0.513, 0.835)	0.689 (0.624, 0.754)	0.658 (0.601, 0.714)	0.596 (0.547, 0.644)	0.568 (0.475, 0.661)
Modified PAGE-B	0.640 (0.601, 0.676)	0.611 (0.577, 0.644)	0.747 (0.592, 0.902)	0.714 (0.650, 0.777)	0.701 (0.648, 0.754)	0.630 (0.581, 0.678)	0.641 (0.550, 0.732)
Modified REACH-B	0.667 (0.630, 0.702)	0.643 (0.606, 0.674)	0.662 (0.516, 0.808)	0.732 (0.670, 0.795)	0.704 (0.654, 0.753)	0.663 (0.616, 0.709)	0.608 (0.515, 0.700)
CAMD	0.565 (0.528, 0.604)	0.553 (0.517, 0.588)	0.674 (0.545, 0.804)	0.626 (0.556, 0.697)	0.603 (0.545, 0.661)	0.553 (0.506, 0.600)	0.576 (0.482, 0.671)
aMAP	0.603 (0.564, 0.641)	0.610 (0.573, 0.645)	0.725 (0.571, 0.879)	0.713 (0.653, 0.774)	0.706 (0.655, 0.758)	0.630 (0.582, 0..678)	0.629 (0.536, 0.722)
HCC-RESCUE	0.588 (0.547, 0.623)	0.560 (0.524, 0.593)	0.761 (0.646, 0.876)	0.692 (0.630, 0.754)	0.645 (0.591, 0.700)	0.571 (0.523, 0.619)	0.513 (0.416, 0.610)
AASL-HCC	0.616 (0.578, 0.655)	0.590 (0.557, 0.623)	0.798 (0.679, 0.916)	0.724 (0.662, 0.787)	0.680 (0.625, 0.735)	0.603 (0.553, 0.653)	0.578 (0.486, 0.670)
Toronto HCC Risk Index	0.603 (0.564, 0.641)	0.572 (0.537, 0.608)	0.693 (0.591, 0.794)	0.709 (0.651, 0.768)	0.667 (0.613, 0.721)	0.589 (0.541, 0.637)	0.536 (0.437, 0.634)
PLAN-B	0.638 (0.600, 0.675)	0.613 (0.578, 0.650)	0.634 (0.489, 0.779)	0.727 (0.658, 0.797)	0.691 (0.638, 0.743)	0.625 (0.576, 0.673)	0.462 (0.365, 0.560)
APA-B (*n* = 910) ^†^	0.661 (0.615, 0.703)	0.655 (0.618, 0.691)	0.608 (0.447, 0.769)	0.651 (0.573, 0.729)	0.664 (0.603, 0.725)	0.679 (0.626, 0.732)	0.718 (0.604, 0.832)
CAGE-B (*n* = 808) ^‡^	0.621 (0.571, 0.675)	0.622 (0.579, 0.661)	-	0.645 (0.533, 0.757)	0.679 (0.606, 0.753)	0.679 (0.606, 0.753)	0.675 (0.556, 0.794)
SAGE-B (*n* = 808) ^‡^	0.639 (0.587, 0.691)	0.637 (0.593, 0.678)	-	0.659 (0.540, 0.777)	0.705 (0.633, 0.777)	0.610 (0.547, 0.674)	0.667 (0.546, 0.786)

* Integrated AUC were calculated up to 8 years after initiating AVT using bootstrap sampling. ^†^ APA-B were calculated for HCC development after 6 months in 910 patients with baseline alpha-fetoprotein result. ^‡^ CAGE-B and SAGE-B were calculated for HCC development after 18 months in 808 patients, with follow-up transient elastography results after 12 months. Abbreviation: CI, confidence interval; AUC, area under the receiver operating characteristic curve; TDAUC, area of the time-dependent receiver operating characteristic curve; HCC, hepatocellular carcinoma.

**Table 5 diagnostics-13-00003-t005:** Cumulative incidence of HCC development in patients with treatment-naïve chronic hepatitis B according to the risk stratification by each risk prediction model.

Risk Stratification	Patient No. (%)	Cumulative Incidence of HCC					Log Rank *p* Value	Log Rank *p* Value vs.	
		1 Year	2 Year	3 Year	5 Year	8 Year			
All Patients	1339 (100)	12 (0.9)	49 (3.8)	91 (7.4)	155 (14.6)	203 (31.7)			
PAGE-B									
Low (0–9)	113 (8.4)	0 (0.0)	0 (0.0)	2 (2.2)	6 (8.5)	10 (24.7)	0.025	Int	0.200
Int (10–17)	951 (71.0)	7 (0.7)	31 (3.4)	57 (6.5)	105 (13.9)	142 (32.3)		High	0.030
High (18–25)	275 (20.5)	5 (1.8)	18 (6.8)	32 (12.9)	44 (19.3)	51 (30.4)		Low	0.020
Modified PAGE-B									
Low (0–8)	120 (9.0)	0 (0.0)	0 (0.0)	2 (2.0)	6 (7.3)	9 (19.4)	<0.001	Int	0.200
Int (9–12)	642 (47.9)	2 (0.3)	13 (2.1)	25 (4.2)	56 (11.4)	79 (24.9)		High	<0.001
High (13–21)	577 (43.1)	10 (1.8)	36 (6.4)	64 (12.1)	93 (19.7)	115 (42.4)		Low	0.002
Modified REACH-B									
Low (0–6)	185 (13.8)	0 (0.0)	0 (0.0)	2 (1.3)	8 (6.7)	12 (19.8)	<0.001	Int	0.020
Int (7–11)	846 (63.2)	8 (1.0)	24 (2.9)	49 (6.3)	86 (12.8)	116 (28.9)		High	<0.001
High (12–16)	308 (23.0)	4 (1.3)	25 (8.4)	40 (14.0)	61 (24.0)	75 (45.1)		Low	<0.001
CAMD									
Low (0–7)	-		-	-	-	-	0.003	-	
Int (8–13)	408 (30.5)	2 (0.5)	8 (2.0)	19 (5.1)	33 (10.2)	44 (21.4)			
High (14–23)	931 (69.5)	10 (1.1)	41 (4.6)	72 (8.4)	122 (16.5)	159 (36.3)			
aMAP									
Low (1–50)	14 (1.0)	0 (0.0)	0 (0.0)	0 (0.0)	0 (0.0)	-	0.003	Int	0.400
Int (50–60)	142 (10.6)	0 (0.0)	1 (0.7)	3 (2.4)	5 (5.1)	8 (14.2)		High	0.002
High (60–100)	1183 (88.3)	12 (1.0)	48 (4.2)	88 (8.1)	150 (15.8)	195 (33.8)		Low	0.100
HCC-RESCUE									
Low ( ≤ 64)	43 (3.2)	0 (0.0)	0 (0.0)	0 (0.0)	1 (3.1)	2 (9.2)	0.070	Int	0.080
Int (65–84)	637 (47.6)	2 (0.3)	11 (1.8)	31 (5.4)	65 (13.5)	93 (33.9)		High	0.200
High (≥85)	659 (49.2)	10 (1.5)	38 (5.9)	60 (9.8)	89 (16.3)	108 (30.8)		Low	0.050
AASL-HCC									
Low (0–5)	-	-	-	-	-	-	0.003	-	
Int (6–19)	763 (57.0)	2 (0.3)	10 (1.6)	29 (4.5)	66 (11.7)	96 (29.0)			
High (20–29)	576 (43.0)	10 (1.8)	27 (6.6)	50 (11.2)	77 (18.4)	95 (35.3)			
Toronto HCC Risk Index									
Low (0–120)	45 (3.4)	0 (0.0)	0 (0.0)	0 (0.0)	0 (0.0)	2 (26.2)	0.030	Int	0.100
Int (120–240)	635 (47.4)	3 (0.5)	9 (1.5)	25 (4.4)	59 (12.4)	90 (32.4)		High	0.050
High (240–366)	659 (49.2)	9 (1.4)	40 (6.3)	66 (10.9)	96 (17.6)	111 (31.5)		Low	0.040
PLAN-B									
Low (0.075–0.250)	190 (20.9)	1 (0.5)	1 (0.5)	2 (1.1)	6 (4.9)	10 (12.1)	<0.001	Int	<0.001
Int (0.250–0.500)	832 (91.4)	6 (0.7)	20 (2.5)	47 (6.2)	95 (14.6)	127 (33.4)		High	0.020
High (0.500–1.000)	317 (34.8)	5 (1.6)	28 (9.2)	42 (14.3)	54 (20.2)	66 (35.8)		Low	<0.001
APA-B (*n* = 910) ^†^									
Low (0–5)	437 (48.0)	2 (0.5)	8 (1.9)	16 (4.0)	27 (7.5)	34 (19.5)	<0.001	Int	<0.001
Int (6–9)	306 (33.6)	4 (1.3)	21 (7.3)	33 (11.9)	50 (20.1)	63 (40.6)		High	0.050
High (10–15)	167 (18.4)	4 (2.5)	11 (6.8)	24 (15.6)	38 (27.3)	50 (66.7)		Low	<0.001
CAGE-B (*n* = 808) ^‡^									
Low	41 (5.1)	-	0 (0.0)	0 (0.0)	1 (3.1)	1 (3.1)	0.001	Int	0.080
Int	468 (57.9)	-	6 (1.3)	17 (3.8)	40 (11.2)	51 (26.8)		High	0.003
High	299 (37.0)	-	6 (2.0)	22 (7.9)	37 (14.9)	56 (45.7)		Low	0.010
SAGE-B (*n* = 808) ^‡^									
Low	198 (24.5)	-	0 (0.0)	1 (0.6)	8 (5.4)	11 (14.0)	<0.001	Int	0.003
Int	450 (55.7)	-	8 (1.8)	24 (5.5)	47 (13.0)	62 (33.7)		High	0.002
High	160 (19.8)	-	4 (2.5)	14 (9.5)	23 (17.8)	35 (50.5)		Low	<0.001

Values are expressed as numbers (percentages). ^†^ APA-B were calculated for HCC development after 6 months in 910 patients with baseline alpha-fetoprotein result. ^‡^ CAGE-B and SAGE-B were calculated for HCC development after 18 months in 808 patients, with follow-up transient elastography results after 12 months. Abbreviation: HCC, hepatocellular carcinoma; Low, low-risk group; Int, intermediate (or median)-risk group; High, high-risk group.

## Data Availability

The data presented in this study are available on request from the corresponding author. The data are not publicly available due to patient privacy concerns.

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
