# Peer review of "Suboptimal Performance of Hepatocellular Carcinoma Prediction Models in Patients with Hepatitis B Virus-Related Cirrhosis"

_diagnostics, 2022, doi:10.3390/diagnostics13010003_

Round 1

Reviewer 1 Report

In the present study, Lee and colleagues compared the accuracy for HCC prediction and the ability to stratify the risk for HCC occurrence in patients with CHB and liver cirrhosis treated with Nuc on long-term FU.

The topic addressed by the authors is clinically relevant and, personally, I really appreciated the decision to include in the study only patients with cirrhosis, that are those at higher risk of HCC development. 

Please find below some specific comments that hopefully will improve the manuscript.

1) To further improve the novelty of the study, the authors are stronlgy encouraged to evaluate the performance of PLAN-B model (PMID: 34606915) and compare its performance to the other scores. 

2) Exclusion criteria: Other significant comorbidieties: please specify.

3) Given the potential value of non-invasive scores for patients startification, it would be usefull to report their accuracy (AUC) in the short-term. The AUC at 3-, 5-, and 8-years are low. Indeed, I would not have expected the score to have moderate-to-good accuracy in the long-term, but a good accuracy in the short time would still be useful in order to identify patients at higher risk of HCC development and that therefore may benefit from a closer monitoring. In this regard, can the authors calculated the 1- and 2-years AUC and add the values in Table 4. Please add also the cumulative incidence at 1 and 2 years of FU (Table 5).

4) Figure 1 and Table 5. How where the low-, medium- and high-risk groups identified? Which cut offs were used? please explain in the text.

however, a moderate to good accuracy in the short term would be usefull to identify patients at higher risk 

Author Response

Response to reviewer’s comments

We were most delighted to learn that our manuscript diagnostics-2017451, entitled “Suboptimal performance of hepatocellular carcinoma prediction models in patients with hepatitis B virus-related cirrhosis” has been subject to opportunity to resubmit for publication in Diagnostics. We have carefully considered the valuable comments and suggestions provided by reviewers, and made efforts to improve the manuscript accordingly. In addition, we reduced the number of citations that became excessive after revision. The followings are point-by-point answers to specific questions raised by the reviewers. We hope that the revised version of manuscript could meet the priority required for the publication.

Reviewer 1

In the present study, Lee and colleagues compared the accuracy for HCC prediction and the ability to stratify the risk for HCC occurrence in patients with CHB and liver cirrhosis treated with NUC on long-term follow-up (FU). The topic addressed by the authors is clinically relevant and, personally, I really appreciated the decision to include in the study only patients with cirrhosis, that are those at higher risk of HCC development. Please find below some specific comments that hopefully will improve the manuscript.

  1. To further improve the novelty of the study, the authors are strongly encouraged to evaluate the performance of PLAN-B model (PMID: 34606915) and compare its performance to the other scores.

Response) Thank you for your keen comment. When we applied the PLAN-B model to our cohort, the c-index and integrated area under the curve (iAUC) were still suboptimal; 0.638 and 0.613, respectively. This is most likely because the cirrhosis itself is also the robust prognostic factor in the PLAN-B model. We additionally provided the performances of the newly analyzed models in Table 4.

  1. Exclusion criteria: Other significant co-morbidieties: please specify.

Response) Thank you for the comment. We amended the sentence to read “6) other significant comorbidities (e.g. end-stage kidney disease, uncontrolled heart failure, pulmonary hypertension, and life-threatening autoimmune disease)”

  1. Given the potential value of non-invasive scores for patients’ stratification, it would be useful to report their accuracy (AUC) in the short-term. The AUC at 3-, 5-, and 8-years are low. Indeed, I would not have expected the score to have moderate-to-good accuracy in the long-term, but a good accuracy in the short time would still be useful in order to identify patients at higher risk of HCC development and that therefore may benefit from a closer monitoring. In this regard, can the authors calculated the 1- and 2-years AUC and add the values in Table 4. Please add also the cumulative incidence at 1 and 2 years of FU (Table 5).

Response) Thank you for the valuable comment. We additionally presented the 1- and 2-year time-dependent AUCs (TDAUCs) cumulative incidences of the models in Table 4 and 5. Since patients with liver cirrhosis have a higher risk of developing HCC than those without cirrhosis, it would be clinically meaningful to confirm the predictive power during short-term follow-up in the cirrhosis cohort. However, the 1-year TDAUC may not be of great clinical significance because it was analyzed excluding patients who developed HCC within 6 months of AVT use.

  1. Figure 1 and Table 5. How where the low-, medium- and high-risk groups identified? Which cut offs were used? please explain in the text.

Response) Thank you for the valuable comment. We set up risk groups according to the risk stratification criteria presented in the papers that had introduced each model, and we presented the previous manuscript. We changed the existing Table S1 to Table S2, and additionally provided the list and the risk stratification with references as Table S1. In addition, we added this information in the Method section (2.3. Calculation of HCC risk scores from prediction models).

  1. However, a moderate to good accuracy in the short term would be useful to identify patients at higher risk.

Response) Thank you for the comment. We agree with this opinion. Even if the c-index or AUC of the various prediction models were not high enough in the cirrhosis cohort, it was confirmed that the risk stratification through the models was sufficiently well performed. However, the patients in the low-risk group with HBV cirrhosis showed a sufficiently high cumulative incidence of HCC (4.9–7.5%/5 years, in models which stratified >10% of patients into the low-risk group) compared to patients in the low-risk group without cirrhosis. Therefore, we wanted to emphasize is that the risk of HCC development in patients with HBV cirrhosis should not be underestimated even they were stratified into the low-risk group according to the prediction models (3rd and 5th paragraph of the revised Discussion section).

Reviewer 2 Report

This retrospective study included 1399 cirrhotic patients who received the first anti-HBV therapy.  Among them 211 (15.1%) patients developed HCC, with a 5-year cumulative incidence of 14.6%. Older age (adjusted hazard ratio [aHR], 1.023), lower platelet count (aHR, 0.997), lower serum albumin level (aHR, 0.578), and greater liver stiffness (aHR, 1.012) were associated with HCC development.  The performance of pre-existing well-validated hepatocellular carcinoma (HCC) prediction models were evaluated in this series.

Comments

1.    The definition of HBV-related cirrhosis by platelet count <150 (109/L) and splenomegaly (>12 cm) may miss a significant number of mild cases and select severe cases.

2.    This is a retrospective study. Please show the missing data rate for each category.

3.    How many patients are diagnosed as liver cirrhosis by pathology or present of varices.

4.    Did the investigators try to search for potential HCC diagnosed in other hospitals?

5.    Please list follow up and treatment duration in Table 2.

6.    In conclusion, pre-existing risk prediction models for patients with HBV-related cirrhosis showed suboptimal predictive performance for HCC development. Please discuss whether this phenomenon is specific to the HBV or also present in the HCV or NASH cases.

7.    Based on the data in this series, what is your new proposal? Was it making any difference for those LSM improved one year after therapy?

8.    For convenience to readers, please list the scoring system with references in a new Table. 

Author Response

Response to reviewer’s comments

We were most delighted to learn that our manuscript diagnostics-2017451, entitled “Suboptimal performance of hepatocellular carcinoma prediction models in patients with hepatitis B virus-related cirrhosis” has been subject to opportunity to resubmit for publication in Diagnostics. We have carefully considered the valuable comments and suggestions provided by reviewers, and made efforts to improve the manuscript accordingly. In addition, we reduced the number of citations that became excessive after revision. The followings are point-by-point answers to specific questions raised by the reviewers. We hope that the revised version of manuscript could meet the priority required for the publication.

Reviewer 2

This retrospective study included 1399 cirrhotic patients who received the first anti-HBV therapy.  Among them 211 (15.1%) patients developed HCC, with a 5-year cumulative incidence of 14.6%. Older age (adjusted hazard ratio [aHR], 1.023), lower platelet count (aHR, 0.997), lower serum albumin level (aHR, 0.578), and greater liver stiffness (aHR, 1.012) were associated with HCC development. The performance of pre-existing well-validated hepatocellular carcinoma (HCC) prediction models were evaluated in this series.

Comments:

  1. The definition of HBV-related cirrhosis by platelet count <150 (109/L) and splenomegaly (>12 cm) may miss a significant number of mild cases and select severe cases.

Response) Thank you for your keen comment. The diagnostic criteria of cirrhosis in the present study from the South Korean guideline (the Korean Association for the Study of the Liver [KASL]). However, we also acknowledge potential bias that a significant number of mild cases had been missed. So, we addressed it as a limitation in the 6th paragraph in the revised Discussion section.

  1. This is a retrospective study. Please show the missing data rate for each category.

Response) Thank you for your keen comment. We additionally described how many patients had missing in the variable (e.g. AFP) and accordingly addressed the valid number of the prediction models in Table 1 (e.g. APA-B, CAGE-B and SAGE-B scores).

  1. How many patients are diagnosed as liver cirrhosis by pathology or present of varices?

Response) Thank you for the comment. Histological diagnosis (including liver biopsy or surgical specimen) was done in 64 patients (4.8%). Esophageal and gastric varices were found in 128 (9.6%) patients.

  1. Did the investigators try to search for potential HCC diagnosed in other hospitals?

Response) Thank you for the keen comment. Since all were treatment-naive patients, patients diagnosed with HCC prior to AVT were completely excluded. HCC development in censored patients could not be confirmed due to the limitations of the retrospective study, and this was outside the scope permitted by the institutional review board.

  1. Please list follow up and treatment duration in Table 2.

Response) Thank you for your comment. We added the follow-up and treatment duration in Table 1 and 2.

  1. In conclusion, pre-existing risk prediction models for patients with HBV-related cirrhosis showed suboptimal predictive performance for HCC development. Please discuss whether this phenomenon is specific to the HBV or also present in the HCV or NASH cases.

Response) Thank you for the comment. Primarily because we focused only on patients with CHB, we cannot clarify whether this phenomenon is specific to the HBV or also present in the HCV or NASH cases. However, theoretically, since “liver cirrhosis” itself had been emphasized as one of the most important prognostic factors in the most HCC prediction models so far and its discriminatory ability must be statistically offset in the cohort with cirrhosis, we cautiously speculate that the similar phenomenon might be observed in patients with other chronic liver diseases. Further studies are required to address this issue. We addressed this point in the 6th paragraph of the revised Discussion section.

  1. Based on the data in this series, what is your new proposal? Was it making any difference for those LSM improved one year after therapy?

Response) Thank you for the critical comment. What we wanted to emphasize is that the risk of HCC development in patients with HB-related cirrhosis should not be underestimated even when they were stratified into the low-risk group according to the prediction models. However, the present study clearly has a limitation in providing new knowledge.

We crudely set the subgroup according to the change of liver stiffness after 1 year (Group A, <13 kPa to <13 kPa, n=492; Group B, <13 kPa to >13 kPa, n=13; Group C, >13 kPa to <13 kPa, n=111; Group D, >13 kPa to >13 kPa, n=192); the 5-year cumulative incidence of HCC was 8.8%, 8.7%, 11.7%, and 20.8% in group A, B, C, and D, respectively (all >1.5%/year). Therefore, even if the liver stiffness was sufficiently stabilized after 1 year of AVT (group C), the risk of HCC development cannot be ignored. Further studies are required to provide the optimized guide among patients with cirrhosis.

  1. For convenience to readers, please list the scoring system with references in a new Table.

Response) Thank you for the valuable comment. We changed the existing Table S1 to Table S2, and additionally provided the list and the risk stratification with references as Table S1. In addition, we added this information in the Method section (2.3. Calculation of HCC risk scores from prediction models).

Reviewer 3 Report

Dr. Jae Seung and colleagues validated existing HCC risk scores in 1,399 patients with HBV-related cirrhosis. The main message is that all these HCC risk scores have suboptimal performance in cirrhotic population, though the original cut-off can still be used for risk stratification. Yet, the low-risk group still has a high HCC risk so the risk stratification may not be so informative. The study is of clinical relevance. My comments are as follows.

Major comments:

1.       Do HCC risk scores that included LSM (modified REACH-B, CAGE-B, SAGE-B) perform better in AUROC than HCC risk scores without LSM (PAGE-B, modified PAGE-B, CAMD)?

2.       Do other HCC risk scores (e.g. aMAP, APA-B, AASL-HCC, HCC-RESCUE, or REAL-B) have similar suboptimal performance in cirrhotic patients?

3.       The Toronto HCC risk index (THRI) is a predictive model to determine the risk of HCC in patients with cirrhosis. Does it perform better than other HCC risk scores that are not specifically derived for cirrhotic patients?

4.       Please consider using a flow chart to present the details of the patients included and excluded. What did other significant comorbidities refer to?

5.       The authors mentioned that “Cirrhosis was diagnosed histologically or clinically as follows: (1) platelet count <150,000/uL and ultrasonographic findings suggestive of cirrhosis, including a blunted, nodular liver edge accompanied by splenomegaly (length > 12 cm), or (2) clinical signs of portal hypertension such as gastroesophageal varices”. Also, in the discussion, they mentioned “Since most of the patients in our study were clinically diagnosed with cirrhosis using ultrasonography and clinical parameters, there might be higher possibilities of over-estimation of cirrhosis, when compared to diagnosis by non-invasive fibrosis tests, such as TE”. If liver stiffness is available now in these patients, why platelet counts but not liver stiffness measurement was used for the definition of cirrhosis? Now in the patients without HCC, the median LSM is relatively low (10.3 kPa), and the 25th percentile is only 6.9 kPa. It is likely that some of these patients did not have cirrhosis.

6.       [Section 2.3] What is FSAC?

7.       CAGE-B and SAGE-B require data at year 5. But now the authors used LS value at 1 year after AVT for the calculation. That seems not a correct use of the two scores. Can the authors comment on that?

8.       What was the time point for calculating the HCC risk scores for patients who started AVT and then developed cirrhosis?

9.       Based on the current result, the authors can confirm the current recommendation of putting all patients with cirrhosis to receive HCC surveillance.

Minor comment:

1.       Cirrhotic patients can have a significant chance of mortality. Now the authors followed the patients for 8 years. Patients were censored when they died or developed other malignant diseases rather than HCC. What is the total proportion of patients who died without HCC or developed other malignant diseases rather than HCC during follow-up? If that is a significant proportion, you may consider using competing risk analysis.

Author Response

Response to reviewer’s comments

We were most delighted to learn that our manuscript diagnostics-2017451, entitled “Suboptimal performance of hepatocellular carcinoma prediction models in patients with hepatitis B virus-related cirrhosis” has been subject to opportunity to resubmit for publication in Diagnostics. We have carefully considered the valuable comments and suggestions provided by reviewers, and made efforts to improve the manuscript accordingly. In addition, we reduced the number of citations that became excessive after revision. The followings are point-by-point answers to specific questions raised by the reviewers. We hope that the revised version of manuscript could meet the priority required for the publication.

Reviewer 3

Dr. Jae Seung and colleagues validated existing HCC risk scores in 1,399 patients with HBV-related cirrhosis. The main message is that all these HCC risk scores have suboptimal performance in cirrhotic population, though the original cut-off can still be used for risk stratification. Yet, the low-risk group still has a high HCC risk so the risk stratification may not be so informative. The study is of clinical relevance. My comments are as follows.

Major comments:

  1. Do HCC risk scores that included LSM (modified REACH-B, CAGE-B, SAGE-B) perform better in AUROC than HCC risk scores without LSM (PAGE-B, modified PAGE-B, CAMD)?

Response) Thank you for the comment. We newly provided the 1- and 2-year TDAUCs in Table 4, and the comparisons of integrated AUC in Table S3. Actually, modified REACH-B, which uses LS value at baseline in the present study, showed the highest c-index among the models, and the integrated AUC was significantly higher than the most of the models not using LS value. However, the models using LS value did not continuously show the higher TDAUCs at 1, 2, and 3 years. Considering that patients with liver cirrhosis are at risk of developing HCC even within a relatively short period of time after follow-up, the superiority of the model cannot be quickly determined even if the c-index or integrated AUC was high in the modified REACH-B. We additionally discussed this in the 5th paragraph of the revised Discussion section.

  1. Do other HCC risk scores (e.g. aMAP, APA-B, AASL-HCC, HCC-RESCUE, or REAL-B) have similar suboptimal performance in cirrhotic patients?

Response) Thank you for the valuable comment. We additionally validated the other risk model as available (APA-B in 910 patients with baseline alpha-fetoprotein results), and provided the results in Table 4 and 5. Conclusively, the suboptimal performance was continuously showed in the newly presented models, and their integrated AUCs were also significantly or tended to be lower than did that of modified REACH-B.

  1. The Toronto HCC risk index (THRI) is a predictive model to determine the risk of HCC in patients with cirrhosis. Does it perform better than other HCC risk scores that are not specifically derived for cirrhotic patients?

Response) Thank you for the comment. THRI also showed suboptimal performance in patients with HBV-cirrhosis. We additionally provided the results in the revised Table 4 and 5.

  1. Please consider using a flow chart to present the details of the patients included and excluded. What did other significant comorbidities refer to?

Response) Thank you for the suggestion. We accordingly revised the inclusion and exclusion criteria and additionally provided the flow chart as Figure S1.

  1. The authors mentioned that “Cirrhosis was diagnosed histologically or clinically as follows: (1) platelet count <150,000/uL and ultrasonographic findings suggestive of cirrhosis, including a blunted, nodular liver edge accompanied by splenomegaly (length > 12 cm), or (2) clinical signs of portal hypertension such as gastroesophageal varices”. Also, in the discussion, they mentioned “Since most of the patients in our study were clinically diagnosed with cirrhosis using ultrasonography and clinical parameters, there might be higher possibilities of over-estimation of cirrhosis, when compared to diagnosis by non-invasive fibrosis tests, such as TE”. If liver stiffness is available now in these patients, why platelet counts but not liver stiffness measurement was used for the definition of cirrhosis? Now in the patients without HCC, the median LSM is relatively low (10.3 kPa), and the 25th percentile is only 6.9 kPa. It is likely that some of these patients did not have cirrhosis.

Response) Thank you for your keen comment. The diagnostic criteria of cirrhosis in the present study are from the South Korean guideline (the Korean Association for the Study of the Liver [KASL]). Even if liver stiffness value is a useful test for predicting liver fibrosis, the clinical diagnosis using platelet count and ultrasonographic findings is a generally accepted method to define cirrhosis. We also acknowledge a potential selection bias and addressed it as a limitation in the 7th paragraph of the revised Discussion section.

  1. [Section 2.3] What is FSAC?

Response) Thank you for the critical comment, and we apologize for causing the confusion. We also previously analyzed the performance of FSAC model using the change of non-invasive fibrosis surrogates after 1 year of antiviral therapy, which had been recently presented by Nam et al. (Am J Gastroenterol. 2021 Aug 1;116(8):1657-1666.). However, we did not have enough data (only for 913 patients), and the performance was also suboptimal (c-index, 0.638 [95% CI, 0.600–0.675]; and iAUC, 0.637 [95% CI, 0.587–0.689]) for predicting HCC development after 18 months of AVT. Therefore, we decided not to present this result, which caused confusion by not removing it from the text.

  1. CAGE-B and SAGE-B require data at year 5. But now the authors used LS value at 1 year after AVT for the calculation. That seems not a correct use of the two scores. Can the authors comment on that?

Response) Thank you for the keen comment. As you commented, CAGE-B and SAGE-B are models that evaluate the risk of developing HCC after 5 years of AVT, when the evolution of hepatic fibrosis is considered to be stabilized. Of course, AVT induces changes in hepatic fibrosis not only for a short period of time but also after long-term treatment. However, analyzing their performance after 5 years of AVT had limitations in that it was difficult to evaluate the short-term risk of cirrhosis patients and compare the predictive performance with the other models. We considered that the improvement of liver stiffness can be sufficiently accompanied even after 1 year of treatment (Am J Gastroenterol 2017, 112, 882-891). Therefore, we calculated the CAGE-B and SAGE-B using the follow-up liver stiffness value by transient elastography after 1 year of AVT, even if this method caused underestimation of the predictive performance of CAGE-B and SAGE-B.

We accordingly revised the Method section, and additionally mentioned the limitation in Discussion section.

  1. What was the time point for calculating the HCC risk scores for patients who started AVT and then developed cirrhosis?

Response) Thank you for the keen comment. The previous analysis made a mistake in not excluding patients who developed HCC prior to 18 months after AVT. CAGE-B and SAGE-B were analyzed for HCC development after 18 months of AVT in patients with HBV-cirrhosis. The index time was the time point of AVT initiation. We revised the Result section and Table 4 and 5. However, this method has the disadvantage that it is difficult to directly compare the performance of each model in the available patient subgroup. Therefore, we newly presented the results of comparison with modified REACH-B, which showed the highest predictive power, through Table S3.

  1. Based on the current result, the authors can confirm the current recommendation of putting all patients with cirrhosis to receive HCC surveillance.

Response) Thank you for your kind comment. An exemption of HCC surveillance among patients without cirrhosis having an extremely low risk of HCC development (<1% at 5 years) by the prediction model would be a way to reduce the socio-economic burden. However, the present study could be meaningful to suggest that even patients stratified into the low risk might have a considerable risk of HCC, once they progressed into cirrhosis.

Minor comment:

  1. Cirrhotic patients can have a significant chance of mortality. Now the authors followed the patients for 8 years. Patients were censored when they died or developed other malignant diseases rather than HCC. What is the total proportion of patients who died without HCC or developed other malignant diseases rather than HCC during follow-up? If that is a significant proportion, you may consider using competing risk analysis.

Response) Thank you for the critical comment. We excluded patients with decompensated liver cirrhosis and with life-threatening comorbidities by the exclusion criteria to rule out the possibility of death from extrahepatic causes as possible. However, there was 30 patients (2.24%) who died during the follow-up period in the present study, and 13 patients (0.97%) died without development of HCC. The median follow-up duration (overall survival) of the 13 patients was 21.9 months after AVT initiation. Therefore, although it cannot be completely ignored, we believe that their statistical effect would be indeed minimal.

Round 2

Reviewer 2 Report

Cirrhosis as a high risk factor for HCC development is well-known.

Please give a table for those 838 patients received LSM 1 year later. Please try to give a cutoff value and compare the HCC incidence between tow groups.

Author Response

  1. Cirrhosis as a high risk factor for HCC development is well-known. Please give a table for those 838 patients received LSM 1 year later. Please try to give a cutoff value and compare the HCC incidence between two groups.

Response) The median value of on-treatment LS value of 838 patients were 8.8 kPa. Patients with on-treatment LS value ≥8.8 kPa had a high risk of HCC development than the others (unadjusted hazard ratio = 2.252 [95% confidence interval, 1.500–3.383], P<0.001). The 2-, 3-, 5-, and 8-year cumulative incidence of HCC development were 1.6%, 3.6%, 11.0%, and 23.7% in patients with on-treatment LS value <8.8 kPa, respectively, and 3.6%, 10.5%, 19.9%, and 56.0% in patients with on-treatment LS value ≥8.8 kPa, respectively (log-rank P<0.001).

   We additionally presented the findings as new supplementary table (Table S4) and paragraph “3.5. On-treatment Ls value in cirrhotic patients with CHB” in the Result section. In addition, we discussed the finding in 3rd paragraph of Discussion section. (“Even patients with an LS value that improved to less than 8.8 kPa after 1 year of AVT also showed a high 5-year cumulative incidence rate (11.0%).”)

Reviewer 3 Report

All the comments have been addressed. Thank you.

Author Response

We sincerely appreciate your efforts to improve our paper.